# More to diverse: Generating diversified responses in a task oriented multimodal dialog system

**Mauajama Firdaus[1]\*, Arunav Pratap Shandeelya[2], Asif Ekbal[1]**

**1** Department of Computer Science and Engineering, Indian Institute of Technology Patna, Bihar, India,
**2** Department of Electrical Engineering, International Institute of Information Technology Bhubaneswar, Orissa, India

\* mauajama.pcs16,asif@iitp.ac.in

## Abstract

Multimodal dialogue system, due to its many-fold applications, has gained much attention to the researchers and developers in recent times. With the release of large-scale multimodal dialog dataset Saha et al. 2018 on the fashion domain, it has been possible to investigate the dialogue systems having both textual and visual modalities. Response generation is an essential aspect of every dialogue system, and making the responses diverse is an important problem. For any goal-oriented conversational agent, the system's responses must be informative, diverse and polite, that may lead to better user experiences. In this paper, we propose an end-to-end neural framework for generating varied responses in a multimodal dialogue setup capturing information from both the text and image. Multimodal encoder with co-attention between the text and image is used for focusing on the different modalities to obtain better contextual information. For effective information sharing across the modalities, we combine the information of text and images using the BLOCK fusion technique that helps in learning an improved multimodal representation. We employ stochastic beam search with Gumble Top K-tricks to achieve diversified responses while preserving the content and politeness in the responses. Experimental results show that our proposed approach performs significantly better compared to the existing and baseline methods in terms of distinct metrics, and thereby generates more diverse responses that are informative, interesting and polite without any loss of information. Empirical evaluation also reveals that images, while used along with the text, improve the efficiency of the model in generating diversified responses.

**Data Availability Statement:** All relevant data are within the manuscript and its Supporting information files.

**Funding:** The research reported in the paper is partially supported by " I MP/2018/002072". All the

## 1 Introduction

Recent advancement in artificial intelligence (AI) has opened new frontiers in conversational agents. Human-machine interaction is an important application of AI that helps humans in their day-to-day lives. Progress in AI has lead to the creation of personal assistants like Apple's Siri, Amazon's Alexa, Microsoft's Cortana that assist humans in their everyday works. The

funders had no role in study design, data collection, and analysis, decision to publish, or preparation of the manuscript. There was no additional external funding received for this study.

**Competing interests:** The authors have declared that no competing interests exist.

capability of the machines to comprehend and complete users' goals has empowered research-ers to build advanced dialogue systems. With the progress in visual question answering [1, 2] and image captioning [3, 4], the use of different modalities in dialogue agents has shown remarkable performance bringing the different areas of computer vision (CV) and natural lan-guage processing (NLP) together. Hence, multimodal dialogue system bridges the gap between vision and language, ensuring interdisciplinary research. Integration of information from dif-ferent modalities, such as text, image, audio and video has known to provide complete infor-mation for building the effective end-to-end dialogue systems [5, 6].

Lately, several works on multimodal dialogue systems [7–9] have encouraged research in this direction by combining information from the different modalities, such as texts, audios, videos and images. Multimodal conversational systems provide completeness to the existing dialogue systems by providing necessary information that lacks in unimodal systems. The visual (in case of images and videos) and audio information help in building robust systems. Dialogue systems are grouped into two broad categories, namely open-domain conversational agents and goal-oriented dialogue systems. Response generation or Natural Language Genera-tion (NLG), which handles the task of presenting the information to the user, is an important aspect of these systems. One of the Conversational AI System's main objectives is to combine language and vision for the development of robust dialogue systems.

## 1.1 Problem definition

The ability to present the information to the user is an important task of every dialogue system. Dialog manager decides on what to say to the user but how to say the information is the sole responsibility of natural language generation (NLG) module of a dialogue system. In our cur-rent work, we focus on response generation in a multimodal setup. In Fig 1, it is evident that the responses are dependent on the visual features as well as the textual information. There-fore, our present work is based on a recently released multimodal dialogue [5] dataset. Our pri-mary focus is to generate diverse and polite responses in a multimodal system by utilizing the information of text and image. The existing systems concentrate on generating the responses, while in this work our target is to make the responses interesting and interactive to increase the customer satisfaction leading to customer retention. The task is challenging as we need to consider the information present in both the images and text, and to make the responses diverse and polite simultaneously. Existing unimodal systems suffer from the problem of gen-erating dull and generic responses like Yes, No, Okay, I don't know etc. The researchers,

U2: Hi there
S2: Hi there! How can I help?
U2: Can you show me some vincent sunglasses?
S2: Similar looking ones are

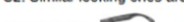
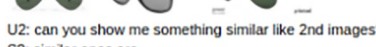

U2: can you show me something similar like 2nd images?
S2: similar ones are

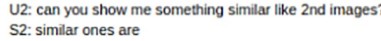

U2: Do you think this looks good with black leather jacket?
S2: No
U2: Any other style collections you have similar to 2nd image?
S2: No

U1: I would like to see a few waist-length Hoodies for myself.
S1: Showing you in just a minute.

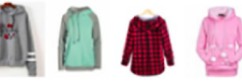

U1: Will gloves complement any of these?
S1: Yes
U1: Do you have the feeling that scarf with fabric type prints will go well with any of these?
S1: Yes
U1: What about the full lengthed leggings?
S1: No.

U1: I would be happy if you can show me resistant toe brogues.
S1: Sorry I don't seem to have anything in metallic but here are some of the different type

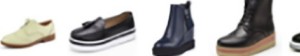

U1: what material is it in the 2nd image?
S1: The brogues in the 2nd image has leather and rubber material
U1: Can I have images of the 4th result from all orientations?

S1:
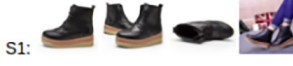

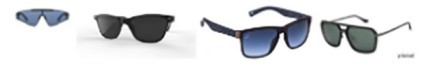

**Fig 1. An example from the MMD dataset.**

therefore, focused on making the responses diverse [10, 11]. Our current work is greatly motivated from these prior works, but our focus is on building a multimodal dialogue system.

The user utterances play a significant role in building the dialogue context for generating coherent and appropriate responses, in accordance to the user demands. We focus on generating the textual responses only in a similar manner as [8, 12, 13]. Here, the task of multi-modal dialog generation is defined as follows: we consider both the modalities, i.e. text and images by effectively combining these two information for the generation of the textual response as opposed to the unimodal systems that typically use only textual information for generating the next response. The primary objective of this work is to generate the diversified and polite responses for increasing the human-machine interaction, and we do not focus on image retrieval or generation in our current work.

## 1.2 Motivation and contribution

Task-oriented dialogue systems are based primarily on textual (unimodal) information. Growing requirements in various fields, such as travel, entertainment, retail etc. require conversational agents to be able to communicate by incorporating information from the different modalities in order to build a robust system. Our research is based on the previously proposed multimodal dialog [5] (MMD) dataset, composed of the conversations related to e-commerce (fashion domain). The work is focused on generating textual responses based on both text and images present in the conversational history. An example from the MMD dataset is shown in Fig 1. It is evident from the figure that the responses are generic and not very interactive. It can be seen that the system's responses are repetitive and mostly in one word like yes or no. Hence, in our current research, we aim at generating the diverse textual responses by capturing information from both text and images. Diversification in textual responses is an open and exciting research problem [10, 14]. This becomes more challenging in multimodal dialogue systems, as information from the images is essential for making the responses diverse while preserving the content as there is a strong correlation between the text and images. For every goal-oriented dialogue systems, the interaction between the user and the system should lead to a better user experience and satisfaction. Hence, the generation of interactive, informative and polite responses is imperative for every conversational agent. Our primary objective is the generation of responses in accordance to the contextual information while making it more diverse and courteous.

In this work, we employ a multimodal hierarchical encoder to encode the information from both text and image. Due to strong dependency between the textual contents and images, we apply parallel co-attention in order to capture the effective evidences from image and text, and to create an useful context for the decoder. We employ BLOCK [15] fusion technique which is based on the block-superdiagonal tensor decomposition for learning better multimodal representation. For increasing the diversity in responses, we apply stochastic beam-search [16] which uses Gumbel Top k-trick for generating responses. To the best of our knowledge, we are the first in presenting a novel approach for making textual responses more diverse in a goal-oriented multimodal dialogue system.

The key characteristics of our present work are as follows:

- We employ a parallel co-attention mechanism to derive the dependencies between the text and image by focusing on the important textual and image information.

- We incorporate BLOCK—Bilinear superdiagonal fusion module to obtain the improved multimodal representations.

- We integrate Stochastic beam search with Gumbel Top k-trick for incorporating diversity in responses while preserving the contextual information.

- We achieve the state-of-the-art performance in making diverse and informative textual responses on the MMD dataset.

## 2 Related work

Response generation is a classical problem in every dialogue systems.Previously, unimodal systems having only text focused on generating responses that are informative, interesting and diverse. With the growth in Artificial Intelligence (AI) systems having images, audio and video modalities have been incorporated in making the robust dialogue systems. Below we present a brief overview of some of the works carried out in unimodal conversational agents followed by the multimodal systems.

### 2.1 Unimodal dialogue systems

Deep learning's efficacy has shown notable improvement in the generation of dialogue. As shown in [17, 18], deep neural models are quite successful in modelling the conversations. To capture the context of previous queries by the user, the authors in [19] proposed a hierarchical framework capable of preserving the past information. Similarly, to preserve the dependencies among the utterances, a hierarchical encoder-decoder framework was investigated in [20, 21]. Sequence to sequence (seq2seq) neural models often generate incomplete and boring responses, such as I don't know, Okay, Yes, No, etc. Hence, bringing the diversity in responses is an extremely challenging and interesting research problem for every conversational agent.

Generation of interesting responses has been the objective in many dialogue generation works, such as [10, 22–27]. In [10], maximum mutual information (MMI) as objective function was proposed for generating the diverse responses in neural models. Similarly, the authors in [23] used inverse token frequency as an objective function for generating the interesting responses. In [24, 25] and [27], conditional variational auto-encoders were used to generate coherent and diversified responses. In [22], adversarial learning was employed for generating informative and diverse responses. Deep reinforcement learning models [26] have also shown remarkable improvement in generating interesting responses. In [28], the inter-sibling ranking penalty was added to favour responses from diverse parents to be generated instead of using the standard beam search algorithm. The authors in [14] proposed diverse beam search algorithm that decodes a list of diverse outputs by optimizing for a diversity-augmented objective.

In [29], a dialogue generation model was proposed that directly captures the variability in possible responses to a given input, which reduces the 'boring/monotonous output' issue of deterministic dialogue models. The generative adversarial network was employed in [30] for generating informative and diversified text. In [31], SpaceFusion model was proposed to jointly optimize diversity and relevance of a sequence-to-sequence model with that of an auto-encoder model by leveraging regularization terms. In [32], the authors proposed a reinforcement learning-based approach which considers a set of responses jointly and generates multiple diverse responses simultaneously. The authors in [11] propose a Frequency-Aware Cross-Entropy (FACE) loss function for generating diverse responses by incorporating a weighting mechanism conditioned on token frequency. In [33], the authors proposed an easy-to-extend learning framework named MEMD (Multi-Encoder to Multi-Decoder), in which an auxiliary encoder and an auxiliary decoder are introduced to provide essential training guidance for generating diverse responses. A multi-mapping mechanism was proposed in [34] to

capture the one-to-many relationship, where multiple mapping modules are employed as latent mechanisms to model the semantic mappings from an input post to its diverse responses.

## 2.2 Multimodal dialogue systems

Recently, research in dialogue systems has shifted towards incorporating different modalities such as images, audio and video for capturing information to make the robust systems. The research reported in [7, 35–38] has been effective in narrowing the gap between vision and language. In [36], an Image Grounded Conversations (IGC) task was proposed, in which natural-sounding conversations were generated about a shared image. Similarly, the authors in [7] introduced the task of Visual Dialog, which required an AI agent to hold a meaningful dialogue with humans in natural, conversational language about visual content. In [39], a combination of generative adversarial networks (GANs) and reinforcement learning was employed to generate more human-like responses to the questions having visual information as well. The authors in [40] addressed the task of cross-modal semantic correlation for Visual dialog that utilized a dual visual attention mechanism for answering the questions. Similarly, for better visual and textual correlation, in [41], a textual-visual Reference-Aware Attention Network was employed for the generation of correct answers in accordance to the input image and dialog history. Our work varies from these as the Multimodal Dialog (MMD) conversations [5] deal with various images, and the growth in conversation depends on both image and text as opposed to a single image conversation.

Recently, video and textual modalities were investigated with the release of the DSTC7 dataset in [9] that made use of a multimodal transformer network to encode videos and incorporate information from the different modalities. Similarly in [6, 42, 43], DSTC7 dataset has been used for generation by incorporating audio and visual features. Earlier works on the MMD dataset reported in [12, 13, 44] used the hierarchical encoder-decoder model to generate responses by capturing information from text, images and the knowledge base. Recently, [8] proposed attribute and position-aware attention for generating textual responses. The authors in [45] used an hierarchical attention mechanism for generating responses on the MMD dataset.

Our work differs from the existing works (mainly, on the MMD dataset) in a sense that we aim here to generate diversified responses using stochastic beam search with Gumble k-tricks at the time of generation for a multimodal dialogue system. The task is challenging as we have to consider the images also for generating the coherent and informative responses. Hence, we focus on capturing vital contextual information by applying co-attention between text and image to achieve important details from both the modalities. We employ BLOCK fusion technique instead of the linear concatenation of modalities to obtain better multimodal representation. The end goal is to achieve responses that are not only coherent to the conversational history but also interesting, diverse and polite.

## 3 Methodology

In this section, we discuss the problem statement followed by the baseline and the proposed methodology.

### 3.1 Formal problem definition

In this paper, we address the task of generating diverse and polite textual responses in accordance to the conversational history in a multimodal dialog setting. The dialogues consist of textual utterance along with the multiple images. Given a context history of $p$ turns, we address

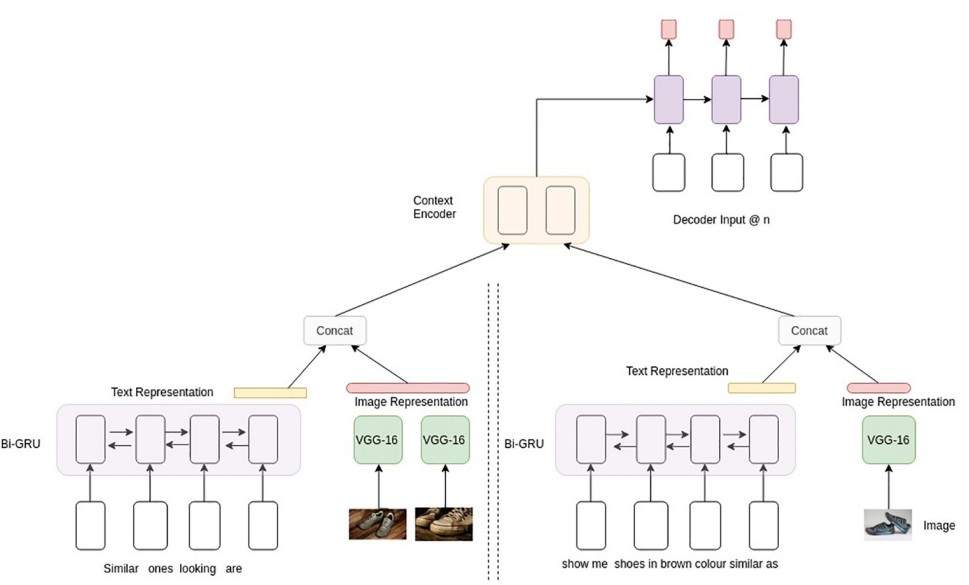

**Fig 2. Architectural diagram of our baseline multimodal hierarchical encoder-decoder.**

the task of generating the next response that is coherent, diverse and polite, leading to better and more engaging human-machine conversation. More precisely, given an user utterance $U_p = u_{p,1}, u_{p,2}, \ldots, u_{p,j}$, a set of images $I_p = img_{p,1}, img_{p,2}, \ldots, img_{p,j'}$, with the dialog history $H_p = (U_1, I_1), (U_2, I_2), \ldots, (U_{p-1}, I_{p-1})$, we focus on generating interesting, informative, polite and context-aware response $Y_p = (y_{p,1}, y_{p,2}, \ldots, y_{p,k})$ instead of template like generic and monotonous responses, such as *I don't know, Yes, No, Similar to...*, etc. This will enhance human-machine conversations by keeping the users engaged in the conversation. Here, $p$ is the $p^{th}$ turn of a given dialogue, while $j$ is the number of words in a given textual utterance and $j'$ is the number of images in a given utterance. Note that in every turn, the number of images $j' \leq 5$, so in-case of only text, vectors of zeros are considered in place of image representation.

## 3.2 Multimodal hierarchical encoder decoder

We construct a generative model for response generation, as shown in Fig 2, which is an extension of recently introduced Hierarchical Encoder-Decoder (HRED) architecture [20, 21]. As opposed to a standard sequence-to-sequence models [46], the dialogue context among the utterances is captured by adding utterance-level RNN (Recurrent Neural Network) over the word-level RNN increasing the efficacy of the encoder to capture the hierarchy in dialogue. The multimodal HRED (MHRED) is built upon the HRED to include text and image information in a single framework. The key components of MHRED are utterance encoder, image encoder, context encoder and decoder.

**3.2.1 Utterance encoder.** Given an utterance $U_p$, we use bidirectional Gated Recurrent Units (BiGRU) [47] to encode each word $n_{p,i}$, where $i \in (1, 2, 3, \ldots..k)$ having $d$-dimensional embedding vectors into the hidden representation $h_{U,p,\ i}$ as follows:

$$\overrightarrow{h_{U,p,i}} = GRU_{U,f}(n_{p,i}, \overrightarrow{h_{U,p,i-1}}) \tag{1}$$

$$\overleftarrow{h_{U,p,i}} = GRU_{U,b}(n_{p,i}, \overleftarrow{h_{U,p,i-1}}) \tag{2}$$

$$h_{U,p,k}^{txt} = [\overrightarrow{h_{U,p,i}}, \overleftarrow{h_{U,p,i}}] \tag{3}$$

**3.2.2 Image encoder.** A pre-trained VGG-16 [48] having a 16-layer deep convolutional neural network (CNN) trained on more than one million of images present in the ImageNet dataset is used for encoding the images. It can classify images into 1000 object categories, such as dress, shoes, animals, keyboard, mouse, etc. As a result, the network can learn rich features from a wide range of images. Here, it is also used to extract the local image representation for all the images in the dialogue turns and concatenate them together. The concatenated image vector is passed through the linear layer to form the global image context representation as given below:

$$T_{p,i} = VGG(I_{p,i}) \tag{4}$$

$$T_p = Concat(T_{p,1}, T_{p,2}, \ldots, T_{p,j'}) \tag{5}$$

$$h_{I,p}^{img} = ReLU(W_I T_p + b_I) \tag{6}$$

where $W_I$ and $b_I$ are the trainable weight matrix and biases. In every turn, the number of images $i \leq 5$, so in-case of only text, vectors of zeros are considered in place of image representation.

**3.2.3 Context encoder.** As shown in Fig 2, the final hidden representations from both image and text encoders are concatenated for each turn, and are given as input to the context level GRU. A hierarchical encoder is built to model the conversational history on top of the image and text encoder. The decoder GRU is initialised by the final hidden state of the context encoder.

$$h_{w,p}^{ctx} = GRU_w([h_{U,p,k}^{txt}; h_{I,p}^{img}], h_{w,p-1}) \tag{7}$$

where $h_{w,p}^{ctx}$ is the final hidden representation of the context for a given turn.

**3.2.4 Decoder.** In the decoding section, we build another GRU for generating the words sequentially based on the hidden state of the context GRU and the previously decoded words. We use input feeding decoding along with the attention [49] mechanism for enhancing the performance of the model. Using the decoder state $h_{d,t}^{dec}$ as the query vector, the attention layer is applied to the hidden state of the context encoder. The context vector and the decoder state are concatenated and used to calculate a final distribution of probability over the output

tokens.

$$h_{q,t}^{dec} = GRU_d(y_{p,t-1}, h_{q,t-1}) \tag{8}$$

$$\alpha_{t,m} = softmax(h_{w,p}^{ctx\,T} W_f h_{q,t}) \tag{9}$$

$$c_t = \sum_{m=1}^{k} \alpha_{t,m} h_{w,p}^{ctx}, \tag{10}$$

$$\tilde{h}_t = tanh(W_{\tilde{h}} [h_{q,t}; c_t]) \tag{11}$$

$$P(y_t/y_{<t}) = softmax(W_S \tilde{h}_t) \tag{12}$$

where, $W_f$, $W_S$ and $W_{\tilde{h}}$ are trainable weight matrices.

## 3.3 Proposed approach

To improve the performance of the MHRED model, we use an attention layer to mask out the insignificant information instead of merely concatenating the representations of the text and image encoder. It is essential to focus on both image and text modalities to achieve informative and coherent responses while generating exciting and diverse responses. For better representation across the modalities, we employ the recently proposed BLOCK fusion technique [15]. We provide the knowledge base information to the decoder for generating informative and context-aware responses. Finally, to generate diverse responses, we incorporate stochastic beam search at the time of generation. Beam search makes use of Gumbel max trick for top-K samples. The architectural diagram of the proposed framework is given in Fig 3.

**3.3.1 Parallel Co-Attention (PCA).** As opposed to visual question answering with single image in general, in multi-modal dialogue system we have multiple images over a context of turns. The MMD dataset has high correlation and dependency between the images and text. To achieve the correct attribute information in accordance to both text and image, it is essential to focus on both the modalities simultaneously. Hence in this work, to generate the context for every turn, we use the parallel co-attention as proposed in [50] to attend over the utterance encoder and image encoder simultaneously. In our case, we apply parallel co-attention between the textual utterance and the multiple images present in the utterance to attain significant attributes for the generation of more informative textual response. In the proposed framework, we connect the images and the utterance by computing the similarity between the utterance and image features at every pair of utterance-locations and image-locations. Since, parallel co-attention draws the essential features among both the text and visual counterparts hence in our work we employ PCA to obtain the important information from the text and the different images present in the input utterance. This facilitates the network in generating attribute centered responses as the attention focuses on the different attributes, such as color, pattern, shape, etc. of the products. The attention network, in accordance to text, captures the appropriate image information from multiple images. For example, the attended image is in consonance to the text such as 3$^{rd}$ image, 2$^{nd}$ image, etc. By achieving the relevant image information the generated response becomes more coherent to the dialogue history. The use of parallel co-attention in dialogue systems is new, especially for multi-modal dialogue setting and under a situation when multiple images exist in a single turn of a dialogue utterance. Precisely, given an utterance representation $U \in \Re^{d \times K}$, image representation $I \in \Re^{d \times N}$, the affinity

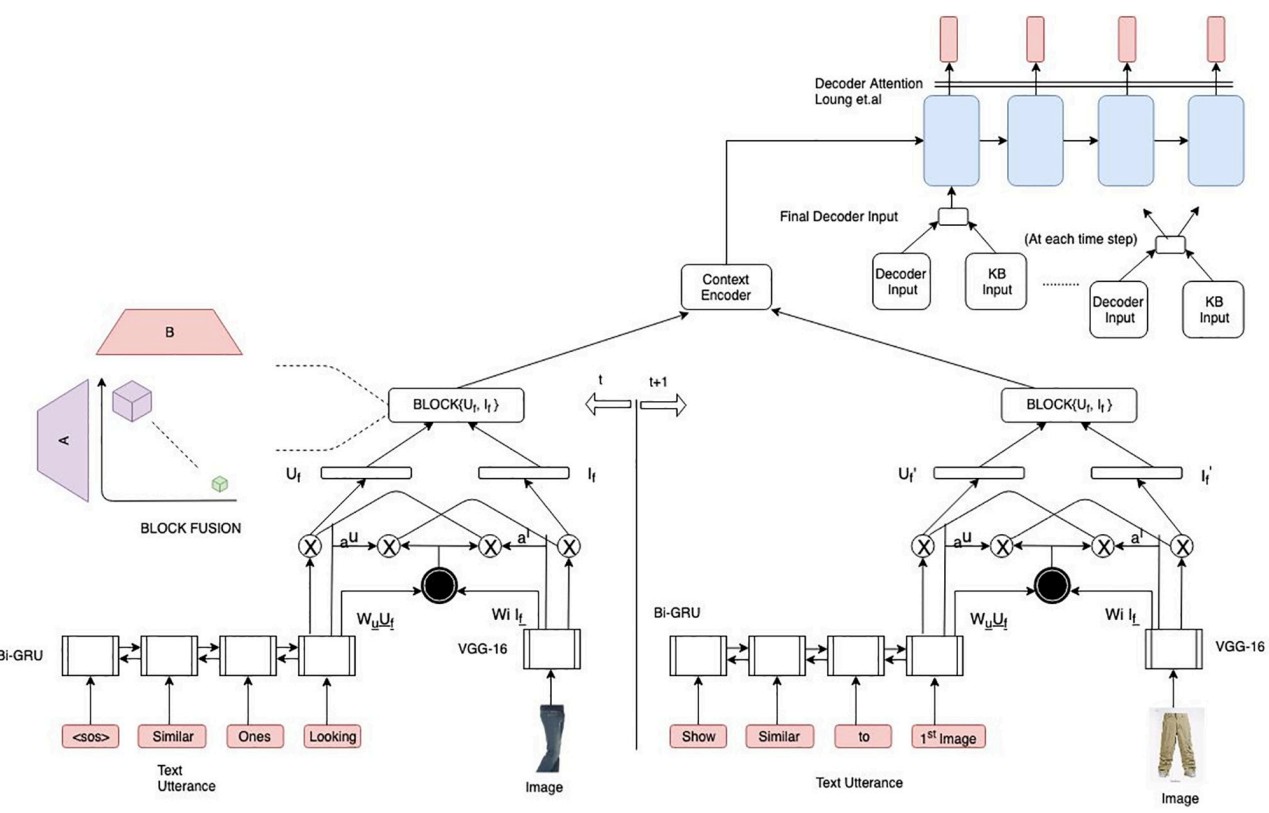

**Fig 3. Architectural diagram of our proposed multi-modal hierarchical encoder-decoder with parallel co-attention mechanism along with BLOCK fusion technique between the text and image.** Input to the decoder is the attended and fused context vector along with the decoder and KB inputs.

matrix $C = \Re^{K \times N}$ can be computed as:

$$C = tanh(U^T, W_b, I) \tag{13}$$

where $W_b$ are the trainable weights. After the computation of the affinity matrix $C$, one of the plausible ways of calculating the utterance or image attention distribution is to simply maximize the affinity over the locations of other modality, i.e. $\alpha^U[n] = max_i(C_{i,n})$ and $\alpha^I[n] = max_j(C_{j,n})$. In contrast to choosing the max activation, the performance can be further improved by considering the affinity matrix $C$ as a feature. Therefore, the attention feature map for parallel co-attention between image and text utterance can be represented as follows:

$$h_I = tanh(W_I I + (W_U U)C) \tag{14}$$

$$h_U = tanh(W_U U + (W_I I)C^T) \tag{15}$$

$$\alpha^I = softmax(W_{h,I} h_I) \tag{16}$$

$$\alpha^U = softmax(W_{h,U} h_U) \tag{17}$$

where $W_{h,I}, W_{h,U} \in \Re^S$ are the weight parameters. Here, $\alpha^I \in \Re^N$ and $\alpha^U \in \Re^K$ are the attention probabilities of each image and word, respectively. The affinity matrix $C$ transforms utterance attention space to image attention space (vice versa for $C^T$). Based on the above attention

weights, the utterance and image vectors are calculated as the weighted sum of the utterance and image features given by:

$$U_a = \sum_{n=1}^{N} \alpha_n^U U_n \tag{18}$$

$$I_a = \sum_{t=1}^{T} \alpha_n^I I_n \tag{19}$$

We employ parallel co-attention between the text utterances and images to obtain a more focused information conditioned on the text utterances. We concatenate $U_a$ and $I_a$, where $U_a$ is the final utterance representation and $I_a$ is the final image representation.

**3.3.2 BLOCK fusion technique.** Multi-modal representation learning is an important aspect of every multi-modal system. One of the very fundamental principles is to fuse the information from the different modalities (e.g. text, audio and visual) effectively for developing a robust dialogue system. In our case, we need better representation to learn the relationships between the textual and visual modalities. Previous works on multi-modal dialogue systems [12, 13, 44, 45] have merely concatenated the textual and visual representation, and thus have only linear interaction between the modalities. This is insufficient in capturing the complex interactions between the diverse modalities. For better non-linear interaction between the two modalities, we apply BLOCK fusion technique. With recent success of BLOCK fusion technique in visual question answering [15], we tend to incorporate it in case of multi-modal dialogue system, where the interaction between the modalities is highly intricate as the previous dialogue history along with multiple images needs to be considered for generating appropriate responses that are contextually coherent with both textual and visual modalities. With the this robust fusion technique, i.e. BLOCK [15], we tend to capture better multi-modal representation. This is based on the concept of block-superdiagonal tensor decomposition. It utilizes the concept of block-term ranks that generalize the notion of both rank and mode-ranks for tensors. It helps in defining new approaches for optimizing the trade-off between the complexity and expressiveness of the fusion model. Hence, it can capture and represent very intricate interactions between the modalities while handling powerful uni-modal representations.

The final image representation $I_a$ and final utterance representation $U_a$ serve as the inputs to the BLOCK module. The BLOCK fusion approach brings state-of-the-art performance for effective interaction between the two modalities, and hence enhances the performance of the system. Therefore, we employ it in our multi-modal encoder to obtain better representation through effective interaction between the textual and visual modalities. In our case, the BLOCK module is computed as:

$$\zeta = C((U_a^T W_U) * (I_a^T W_I)) \tag{20}$$

where $W_U$ and $W_I$ are the trainable parameters, and $*$ denotes the element-wise product.

Hence, the final multi-modal BLOCK fusion between the final textual representation, final image representation and contextual representation is computed as:

$$h_d = BLOCK(U_a, I_a) \tag{21}$$

**3.3.3 Knowledge base (KB).** The knowledge base encoder used in our framework is the same as [13]. The knowledge base of the MMD dataset contains information about the contextual queries and celebrities endorsing various products and brands. Hence, to provide this

additional information to our proposed model, we employ self-attention on KB input to achieve more focused information as follows:

$$h_s^{query} = n_p^{query}(h_{n-1}^{query}, k_{l,t}) \tag{22}$$

$$h_f^{ent} = n_p^{ent}(h_{n-1}^{ent}, d_{l,t}) \tag{23}$$

$$h_{net}^{kb} = [h_s^{query}, h_f^{ent}] \tag{24}$$

$$\beta = softmax(W_{x1} tanh(W_{v2} h_{net}^{kb})) \tag{25}$$

where $W_{x1}$ and $W_{v2}$ are trainable parameters.

We use the attended KB output along with the decoder input as the combined input at each time step of the decoder.

As the knowledge base (KB) input remains intact for a particular dialogue context, therefore we concatenate the KB input with the decoder input in a similar manner as in [13]. Apart from that, we fuse the text and visual representation at the encoder level to form the context for every dialogue turn. During decoding, to make the responses diverse, we use stochastic beam search [16] for diversification and informative responses in accordance to the contextual information.

$$h_{d,t}^{dec} = f^{dec}(h_{d,t-1}^{dec}, W_{d,t}, h_{w,p}^{ctx}, h_{net}^{kb}) \tag{26}$$

## 3.4 Training and inference

The whole model is trained using teacher-forced cross entropy [51] at every decoding step to minimize the negative log likelihood on the model distribution. We define $\hat{y} = \hat{y}_1, \hat{y}_2, \hat{y}_3, \ldots, \hat{y}_m$ as the ground truth of the given input sequence.

$$J(\theta) = -\sum_{s=1}^{n} logp(\hat{y}_t | y_1, y_2, \ldots, y_{m-1}) \tag{27}$$

For diversity, we use Stochastic beam search for generating the response, which is more diversified and also capable of preserving the contextual information. Stochastic beam search was derived from the Gumbel Top-K trick to sample sequences without replacement. This requires instantiating all the sequences in the domain to get the largest perturbed probability. Then we transit to a top-down sampling of the perturbed log probabilities.

**3.4.1 Gumbel top K tricks.** As mentioned in [16], the model is represented as a tree where internal nodes at level $p$ represent the partial sequences $y_{1:p}$ and the nodes corresponding to leaf nodes are the completed sequences $y^i$. We identify the leaf nodes by its index $i \in (1, 2, \ldots n)$ and write $y^i$ as the corresponding sequence with log probability $\phi_i = logp_\theta(y^i)$. The Gumbel max-trick samples from this distribution by independently perturbing the log probabilities with Gumbel noise and finding the largest element. Now, this generalizes the Gumbel Max Trick to Gumbel Top-K tricks to sample with the size K without any replacement, by taking the indices of k-largest perturbed log-probabilities.

## 3.5 Baseline models

For our experiment, we implement the following baseline models.

**Model 1 (MHRED):** The first model is the baseline MHRED model as previously described in the methodology section. This is shown in Fig 2, that has utterance encoder, image encoder, context encoder and the decoder for generating the responses.

**Model 2 (MHRED+KB):** In this model, we add the KB information at the decoder to incorporate contextual queries and celebrity information for endorsing different brands and products for informative response generation that is in accordance with the conversational history.

**Model 3 (MHRED+KB+A):** In this model, we employ global attention [49] at the decoder for better decoding at each time step.

**Model 4 (MHRED+KB+A+DAA(I,T)):** In this model, for different attributes on the text, we implement dynamic attribute attention to obtain the enhanced representation computed by:

$$\beta_t = softmax(W_p^T H_u) \tag{28}$$

$$U_p = \beta_t H_u^T \tag{29}$$

Here, a self-attended text embedding is used as query $U_p$ to calculate the attention distribution over the image feature representation as $H_I = [H_{I,1}, H_{I,2}, H_{I,3}, \ldots, H_{I,j'}]$.

$$\beta_k = softmax(U_p^T W_{p'} H_I) \tag{30}$$

$$I_a = \beta_k H_I^T \tag{31}$$

**Model 5 (MHRED+KB+A+PCA(I,T)):** In this model, we employ parallel co-attention mechanism as described in 3.3.1 to attend the image and text simultaneously. We connect image and text by calculating the similarity between the image and text features at all the pairs of image and text utterances, as previously described in the methodology section.

**Model 6 (MHRED+KB+A+PCA(I,T)+MFB(I,T)):** In this model, we concatenate pairwise text and image features $U_f$ and $I_f$ obtained after parallel co-attention mechanism as input to the MFB module for better interaction between the modalities as used in [8].

**Model 7 (MHRED+KB+A+PCA(I,T)+BLOCK(I,T)):** In this model, we concatenate the pairwise output of text and image features $U_a$ and $I_a$ obtained after parallel co-attention mechanism as input to the BLOCK module where the output of BLOCK serves as input to the context encoder.

## 4 Dataset

Our research is based on the Multi-modal Dialog (MMD) dataset [5] consisting of 150k chat sessions between the customer and sales representative. During the sequence of customer-agent interactions, domain-specific information in the fashion domain was collected. The dialogues easily integrate text and image knowledge into a conversation that brings together different modalities to create a sophisticated dialog system. The dataset presents new challenges for multi-modal, goal-oriented dialogue systems having complex user sentences. The detailed information of the MMD dataset is presented in Table 1. The authors [5], for experimentation unroll the different images to incorporate only one image for a single utterance. This method, though computationally learns, eventually lacks the goal of capturing multi-modality over the context of multiple images and text. Therefore, in our study, we use a different version of the dataset as outlined in [12, 13] to capture a large number of images as the concatenated context vector for each turn of a dialogue. The motivation behind this is the fact that multiple images are required for providing the correct responses to the users. As shown in the example before,

**Table 1. Dataset statistics of multi-modal dialogue (MMD) dataset.**

| Dataset Statistics | Train | Valid | Test |
|---|---|---|---|
| *Number of dialogues* | 105,439 | 22,595 | 22,595 |
| *Avg. turns per Dialogue* | 40 | 40 | 40 |
| *No. of Utterances with Text Response* | 1.54M | 331K | 330K |
| *Avg. words in Text Response* | 14 | 14 | 14 |
| *No. of Utterances with Image Response* | 904K | 194K | 193K |

can you show me something similar like the 2nd image?, the different images present play an important role for the generation of contextually correct responses.

For incorporating diversity and politeness in the generated responses, we manually annotated 40% of the training set dialogues with courteous phrases and varied responses that are contextually coherent with the dialogue history and the product being discussed in the conversation. Three annotators proficient in English language were assigned to annotate the textual responses making it more empathetic and diverse by incorporating phrases for apology, appreciation, assurance and greetings. We observe the multi-rater Kappa agreement ratio of approximately 80%, which may be considered as reliable. The annotated courteous data was used for training the model for making it more polite and diverse. An example of the polite version of the MMD dataset annotated by experts is shown in Fig 4.

## 5 Experiments

In this section, we present the information about the implementation details of our proposed framework.

### 5.1 Implementation details

All the implementations have been performed using PyTorch https://pytorch.org/ framework. We use 512-dimensional word embedding initialized randomly. We did not use any delexicalisation, and our model learns independent of the context encoder and knowledge base (KB). All the encoder and decoder have 1-layer GRU cell with 512 hidden dimensions. For image representation, we use 4096-dimensional FC6 layer from VGG-16 [48], pre-trained on the Imagenet. We use AMSGrad [52] as the model training optimizer to alleviate the problems of slow convergence. We use dropout [53] with the probability of 0.4 to avoid over-fitting. All the parameters are randomly initialized using Gaussian distribution with Xavier scheme [54]. For generating diversified responses, we use the different methods, such as beam search, diverse beam search and greedy sampling. Finally, in our proposed model, we employ stochastic beam search with sample size k: 5, 10 as implemented in fairseq https://ai.facebook.com/tools/fairseq/. It works based on the Gumbel Top-k trick that samples sequences without replacement from the sequence model. We experiment with different learning rates and lastly fix it as 0.0004.

To measure the politeness quotient in the generated response we design a politeness classifier as presented in Fig 5. The input is first converted into embeddings using the embedding layer. We use 300-dimensional Glove embedding for representing the utterances. The embedded utterance is fed to the convolutional layer with filter size 3. After convolution, we apply max pooling to obtain the hidden representations. For more abstract representation of the dialogue utterance, we apply a unidirectional LSTM network on the hidden representation followed by self-attention. The number of neurons on the LSTM layer is fixed to be 200. Finally, we apply the softmax layer for attaining the politeness accuracy of the given utterance. To

U1: Hi there!
S1: Hello! how I can help you?
U1: Can you show me some trousers for my father-in-law.
S1: The similar one looking are

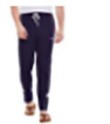 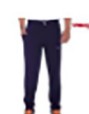 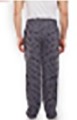 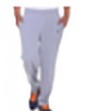 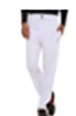
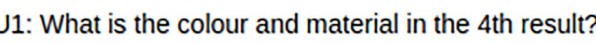

U1: What is the colour and material in the 4th result?
S1: It has grey colour cotton material.
U1: Is the 3rd image good for dry cleaning?
S1: No.

U1: I would like to see a few waist-length Hoodies for myself.
S1: Showing you in just a minute.

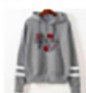 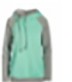 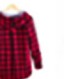 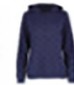

U1: Will gloves complement any of these?
S1: Yes
U1: Do you have the feeling that scarf with fabric type prints will go well with any of these?
S1: Yes
U1: What about the full lengthed leggings?
S1: No.

**Orininal MMD dataset**

U1: Hi there!
S1: Hello please tell me, anything that I can help you with?
U1: Can you show me some trousers for my father-in-law.
S1: Sure, thanks for sharing the information, here are some images for your reference

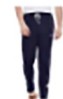 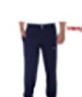 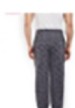 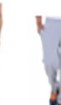 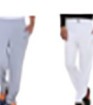
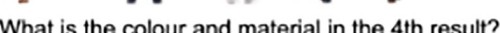

U1: What is the colour and material in the 4th result?
S1: The trackpants in the 4th image has grey colour cotton material.
U1: Is the 3rd image good for dry cleaning.
S1: Sorry, but we won't suggest the 3rd image for dry clean.

U1: I would like to see a few waist-length Hoodies for myself.
S1: Please have patience, we are showing you in just a minute.

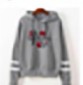 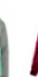 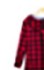 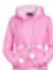 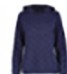

U1: Will gloves complement any of these?
S1: Yes they will suit the hoodies really well.
U1: Do you have the feeling that scarf with fabric type prints will go well with any of these?
S1: Yes printed scarfs will be well suited with these.
U1: What about the full lengthed leggings?
S1: I am sorry but I don't think it will be good with the hoodies.

**Polite and Diversified MMD dataset**

**Fig 4. An example of the new diversified and polite version of the MMD dataset.**

avoid over-fitting of the neural network a useful regularisation technique known as dropout has been used in our model. At the time of forward propagation, the neurons are randomly tuned-off so that the convergence of weights is restricted to the identical positions. For optimisation and regularisation, we use Adam optimiser along with 15% dropout in our model. Categorical cross-entropy is employed to update the model parameters.

## 6 Results and discussion

In this section, we present the evaluation metrics (automatic and human), report the experimental results along with the detailed analysis and comparisons.

### 6.1 Evaluation metrics

**6.1.1 Automatic evaluation metrics.** To evaluate the model at relevance and grammatical level, we report the results using the standard metrics like:

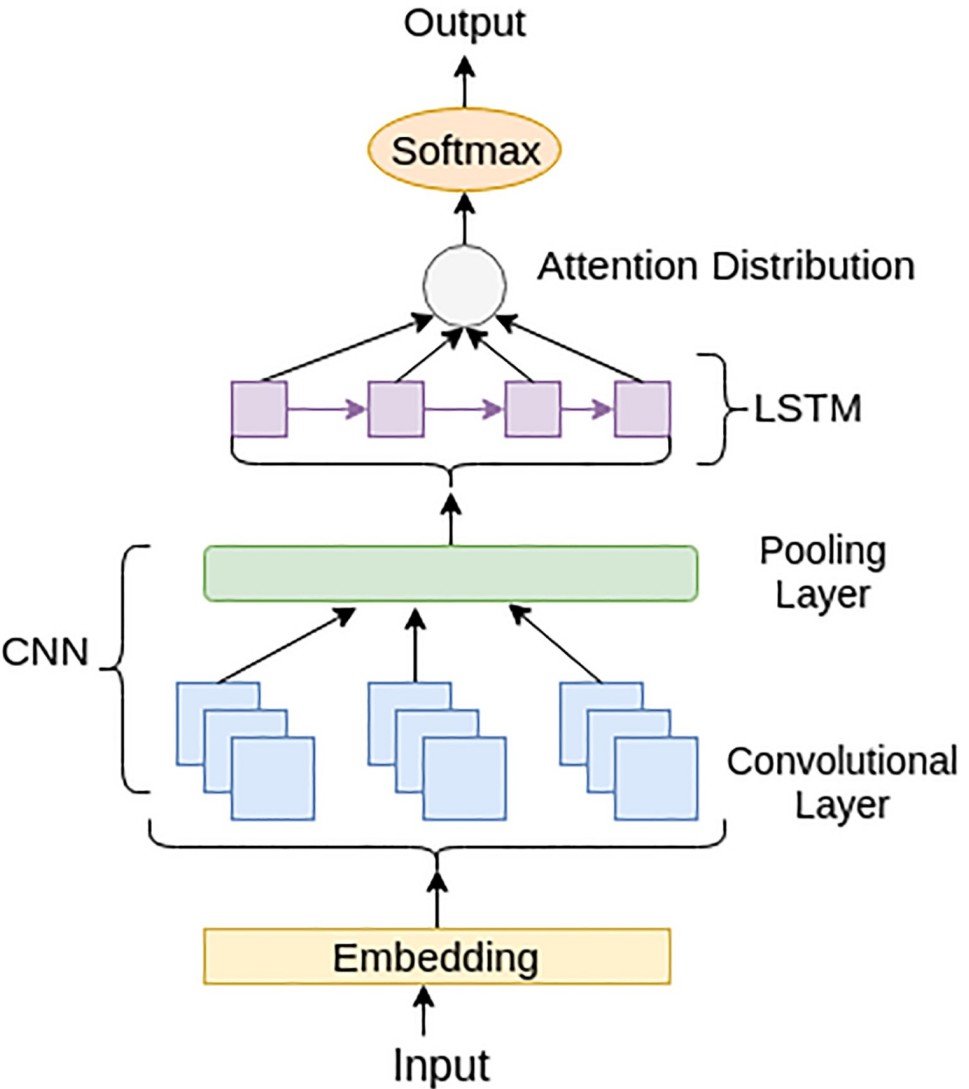

**Fig 5. Architectural diagram of politeness classifier.**

- Perplexity [55]: Perplexity has been used to test our model at the content level. Smaller scores of perplexity mean the responses generated are more grammatical and fluent.

- BLEU-4 [56]: BLEU measures the n-grams overlap between the generated response and the gold response, and has become a common measure for comparing task-oriented dialog systems. It is used to measure the content preservation in the generated responses. BLEU is weakly correlated with human judgments [57], but a low BLEU score across different models suggests that a dataset is highly complex.

- Distinct-1 and Distinct-2: We report the degree of diversity by calculating the number of distinct unigrams (distinct-1) and bigrams (distinct-2) in the generated responses. The resulting metric is, thus, a type-token ratio for unigrams and bigrams as mentioned in [10]. This metric measures the number of distinct k-grams in the replies generated and is scaled by the total number of tokens generated to avoid long replies. This metric is an indicator of word-level diversity for the generated responses.

- Politeness Accuracy: We also compute politeness score using a pre-trained classifier as shown in Fig 5 (trained on Stanford Politeness corpus) for measuring the degree of politeness in the generated responses similar to [58]. The classifier takes as input the generated response and predicts a probability value giving us the politeness accuracy of the generated response.

**6.1.2 Human evaluation metrics.** We adopt human evaluation metrics to compare the efficiency of distinct models in order to examine the quality of the generated responses. We randomly sample 1500 responses from the test set for human evaluation. Three human annotators with post-graduate exposure were assigned to evaluate the generated response in terms of the various human evaluation metrics as mentioned below. Given an utterance, images along with the conversation history, annotators were asked to evaluate the correctness, polite preservation, diversity and relevance of the responses generated by the different models.

- Fluency (F): The generated response is grammatically correct and is free of any errors.

- Diversity (D): It is used to judge whether the generated response is diverse in comparison to the ground truth.

- Relevance (R): The response generated is focused on the aspect being discussed (style, colour, material, etc.) and is in accordance with the conversational history.

- Politeness (P): The generated responses are polite and in accordance to the context having both text and images.

We follow the scoring scheme for fluency and relevance as- 0: incorrect or incomplete, 1: moderately correct, and 2: correct. The scoring scheme for diversity and politeness preservation is 0: for the absence of diversity and politeness in the reply and 1: for the presence of diversity and politeness in the response. We compute the Fleiss' kappa [59] for the above metrics to measure inter-rater consistency. The kappa score for fluency and relevance is 0.75, and for diversity and politeness 0.77, indicating substantial agreement.

## 6.2 Results of automatic evaluation

We present the results of our experiments in Table 2. It is evident from the table that our proposed approach outperforms the baselines in generating diversified responses, and these improvements are statistically significant We perform statistical significance t-test [60] and it is conducted at 5% (0.05) significance level. The perplexity score of the proposed approach with k = 5 is the least, thereby proving the fact that the generated responses are better from the rest. This suggests that our proposed framework employing stochastic beam search with gumble top k-tricks(here k = 5) is able to generate better responses in contrast to the baseline models and the existing system. In the baseline approaches, the framework employing PCA and BLOCK fusion technique outperforms all the other baseline networks. This indicates that both PCA and BLOCK fusion employed together makes the network perform better as opposed to the frameworks in which they are employed individually. Similarly, there is a decrease in perplexity of about 15 points when we use BLOCK instead of MFB fusion technique that states the superiority of BLOCK fusion in comparison to the MFB fusion technique.

Similarly, the distinct-1 and distinct-2 metrics of the final model with stochastic beam search decoding proves that the proposed framework have been successful in generating diverse responses for the MMD dataset. Our proposed framework outperforms the existing approaches such as MMI [10], MMI-antiLM [10] and MMI-Bidi [10] for both distinct-1 and

**Table 2. Results of different models on MMD dataset.**

| | Model Description | Perplexity (PPL) | BLEU-4 | Distinct-1 (d-1) | Distinct-2 (d-2) | Politeness Accuracy (PA) |
|---|---|---|---|---|---|---|
| *Existing Approaches* | *Agarwal et. al.* [13], *(MHRED + Attn)* | 1.098 | 0.4451 | - | - | - |
| | *Chauhan et. al.* [8], *(MHRED + Attn)* | 1.082 | 0.4454 | - | - | - |
| | *Li et. al.* [10], *(MMI)* | 1.789 | - | 0.0315 | 0.0865 | - |
| | *Li et. al.* [10], *(MMI—AntiLM)* | 1.980 | - | 0.0201 | 0.0906 | - |
| | *Li et. al.* [10], *(MMI-bidi)* | 1.967 | - | 0.0244 | 0.0954 | - |
| *Baseline Approaches* | *MHRED* | 1.009 | 0.4561 | 0.0384 | 0.1201 | 0.76 |
| | *MHRED+KB* | 1.009 | 0.4591 | 0.0435 | 0.1595 | 0.76 |
| | *MHRED+KB+A* | 1.008 | 0.4624 | 0.0405 | 0.1453 | 0.77 |
| | *MHRED+KB+A+DAA(I,T)* | 1.005 | 0.4715 | 0.0352 | 0.1205 | 0.78 |
| | *MHRED+KB+A+PCA(I,T)* | 1.004 | 0.4875 | 0.0429 | 0.1138 | 0.78 |
| | *MHRED+KB+A+PCA(I,T)+MFB(I,T)* | 1.0036 | 0.4965 | 0.0358 | 0.0981 | 0.78 |
| | *MHRED+KB+A+PCA(I,T)+BLOCK(I,T)* | 1.0021 | **0.4981** | 0.0399 | 0.1089 | 0.80 |
| *Proposed Approaches* | *MHRED+KB+A+PCA(I,T)+BLOCK(I,T)+SBS(K = 5)* | **1.0008** | 0.4691 | **0.0547** | **0.1653** | **0.84** |
| | *MHRED+KB+A+PCA(I,T)+BLOCK(I,T)+SBS(K = 10)* | **1.0009** | 0.4359 | **0.0501** | **0.1393** | **0.81** |

distinct-2 metrics. From the baseline models, there is an improvement in the distinct-1 and distinct-2 measures for the proposed approach having stochastic beam search decoder for both k = 5 and 10. This implies the efficacy of the stochastic beam search that ensures diverseness in the responses. The distinct-1 and distinct-2 scores demonstrate a huge jump in comparison to the best performing baseline network *MHRED+KB+A+PCA(I,T)+BLOCK(I,T)*, thereby illustrating the effectiveness of stochastic beam search with Gumble k-tricks. From the table it is visible that our baseline framework shows improved performance in case of BLEU-4 metric in comparison to the existing systems. We observe that there is a drop in BLEU-4 metric in our proposed model,*MHRED+KB+A+PCA(I,T)+BLOCK(I,T)* compared to the baseline. This favors our assumption that the generated response is diverse in nature, and hence is not completely similar to the ground-truth. The politeness score also increases in our proposed framework with an increment of 4% with respect to the final baseline model. This improvement indicates that the generated responses are more polite than the baselines. Therefore, our proposed approach ensures that in the multi-modal setup, both diversity and politeness are preserved.

## 6.3 Analysis of network parameters

We provide the analysis of different hyper-parameters such as dropout and learning rate for our proposed framework. We used Perplexity (PPL) as the primary metric to fine-tune the framework for determining the network parameters in case of dropout and learning rate. After fixing these parameters we do the complete evaluation in terms of both automatic and human metrics for all the baselines and proposed framework. We used politeness accuracy (PA) as the primary metric for determining the number of LSTM layers in the politeness classifier. The dropout probability was determined to be 0.4 by considering the range of 0.1—0.8. The statistical analysis of the results for the different values of dropout probability for the proposed framework is provided in Table 3 of the revised manuscript. In Fig 6(a) we depict the performance of the proposed framework in case of dropout. It is evident that the proposed framework performs best when the dropout probability is 0.4. Therefore, for all the experiments (all baselines)

**Table 3. Analysis on proposed framework for learning rate and dropout.**

| Proposed Framework | MHRED+KB+A+PCA(I,T)+BLOCK(I,T)+SBS(K = 5) | | | | | | | |
|---|---|---|---|---|---|---|---|---|
| *Learning Rate* | 0.0010 | 0.0008 | 0.0006 | 0.0005 | 0.0004 | 0.0003 | 0.0002 | 0.0001 |
| *PPL* | 1.0036 | 1.0021 | 1.0019 | 1.0013 | **1.0008** | 1.0017 | 1.0022 | 1.0020 |
| *Dropout* | 0.1 | 0.2 | 0.3 | 0.4 | 0.5 | 0.6 | 0.7 | 0.8 |
| *PPL* | 1.0054 | 1.0031 | 1.0015 | **1.0008** | 1.0011 | 1.0013 | 1.0014 | 1.0012 |

and metrics we fix the dropout probability as stated. For learning rate we determined the value to be 0.0004 similarly as [12, 13, 45]. Also, we cross-verified by taking a range of 0.001 to 0.0001 to determine the learning rate in case of our proposed framework. As shown in Table 3 and Fig 6(b), the proposed framework performs best when the learning rate is fixed at 0.0004 in a similar manner as the existing literature.

For the politeness classifier, we had fixed the LSTM layers at 200 by evaluating the performance of the classifier for the responses generated by our proposed framework. We checked the performance of the classifier by using different layers of LSTM as shown in Table 4 of the

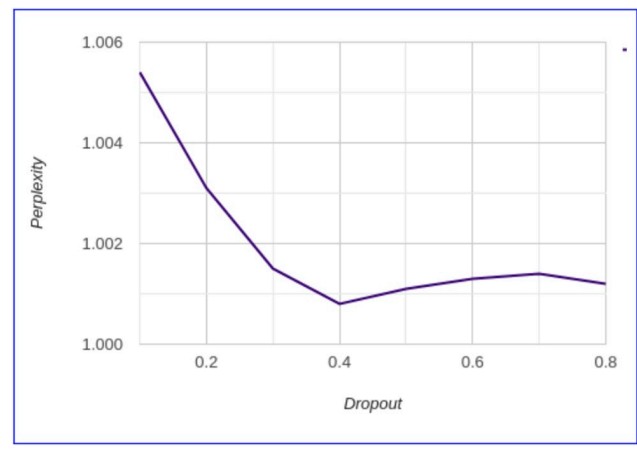

**(a)** PPL vs Dropout

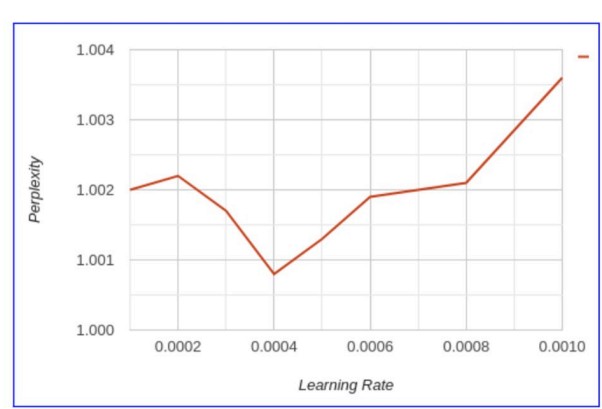

**(b)** PPL vs Learning Rate

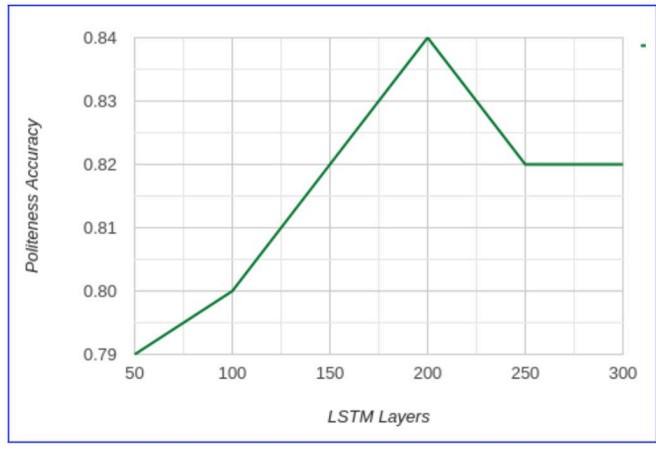

**(c)** PA vs LSTM Layers

**Fig 6. Analysis of different network parameters on the proposed framework.**

**Table 4. Analysis of LSTM layers on politeness classifier for the proposed framework.**

| Proposed Framework | MHRED+KB+A+PCA(I,T)+BLOCK(I,T)+SBS(K = 5) | | | | | |
|---|---|---|---|---|---|---|
| *LSTM Layers* | 50 | 100 | 150 | 200 | 250 | 300 |
| *Politeness Accuracy* | 0.79 | 0.80 | 0.82 | **0.84** | 0.82 | 0.82 |

revised manuscript. From the Fig 6(c), it is evident that by increasing the number of LSTM layers the performance of the classifier improved with the maximum accuracy being 0.84 in case of 200 layers. On increasing the number of layers the performance declined and eventually became constant with accuracy lesser than the best performing classifier at 200 layers. This analysis helped in deciding the number of lSTM layers for the classifier.

## 6.4 Comparisons to the existing systems

In Table 2, we provide the results of our proposed framework in comparison to the existing state-of-the-art methods. For response generation in multi-modal systems, we compare our current work with the existing systems [8, 13]. For a fair comparison with the existing systems, the data used for experimentation should be similar in terms of its structure, genre as well as annotation in order to draw correct conclusions on the results obtained. Hence, we compare our proposed approach with these systems as the dataset is identical. Also, both the existing systems used for comparing with our current system focuses on the task of textual response generation in multi-modal dialogue systems. In [13], the authors employed a knowledge base for generating the textual responses while using both the textual and visual information. Similarly, in [8], the authors have exploited the position and attribute information in text and images for generating the textual responses. As it is evident from Table 2, our proposed system outperforms the existing systems significantly in terms of Perplexity and BLEU-4 metrics. By applying parallel co-attention and BLOCK fusion technique, our best performing baseline framework achieves an improvement of around 5% in BLEU score from both the existing approaches. This is mainly due to the fact that the parallel co-attention network simultaneously captures the essential information from the entire utterance possessing textual and visual knowledge. We also attain enhanced representation of the utterance by having improved interaction between the modalities with the help of BLOCK fusion technique. One of the purposes of the current research is to accomplish diverse response generation. Evaluation shows that by applying stochastic beam search, we observe an improvement of 2% in BLEU score (with k = 5) as opposed to [8] and [13]. The increase in BLEU score signifies that the responses are diverse and different from the ground-truth responses, thereby, helping in achieving the desired task. Also, the decrease in perplexity score testifies that the generated responses are grammatically correct and better in comparison to all the baselines and the existing frameworks. The other existing systems on multi-modal dialog systems focus on the tasks of response generation and image retrieval both, which does not come within the scope of our work. Moreover, the datasets used in these works are different and therefore correct conclusions cannot be drawn for fair comparisons.

It is to be noted that as this is the very first work that focuses on generating diverse responses in a multi-modal setup, hence we are unable to compare our work with any other systems with respect to the diversity. Since the primary goal of this research is to bring diverseness in responses, therefore we compare our proposed system with the state-of-the-art techniques [10] for generating diverse responses. In the existing methods the authors employed various objective functions, such as maximum mutual information (MMI), anti language models (anti-LM), MMI-bidi approach for making the responses diverse. We analyze the

**Table 5. Results of human evaluation on different models for the MMD dataset.**

| | Model Description | Fluency | | | Relevance | | | Diversity | | Politeness Preservation | |
|---|---|---|---|---|---|---|---|---|---|---|---|
| | | 0 | 1 | 2 | 0 | 1 | 2 | 0 | 1 | 0 | 1 |
| *Baseline* | *MHRED* | 25.62 | 38.54 | 35.84 | 42.14 | 35.98 | 21.88 | 57.13 | 42.87 | 44.73 | 55.27 |
| | *MHRED+KB+A+PCA(I,T)+BLOCK(I,T)* | 22.14 | 40.22 | 37.64 | 31.66 | 38.45 | 29.89 | 54.22 | 45.78 | 41.69 | 58.31 |
| *Proposed* | *MHRED+KB+A+PCA(I,T)+BLOCK(I,T)+SBS(K = 5)* | 13.81 | 44.88 | 41.31 | 26.59 | 38.54 | 34.87 | 25.68 | 74.32 | 38.07 | 61.93 |
| | *MHRED+KB+A+PCA(I,T)+BLOCK(I,T)+SBS(K = 10)* | 14.35 | 45.17 | 40.48 | 28.37 | 37.32 | 34.31 | 27.84 | 72.16 | 39.77 | 60.23 |

responses generated by these techniques on the MMD dataset to have a thorough comparison in terms of distinct-1 and distinct-2 metrics. From the table, it is visible that our proposed framework performs better than the existing approaches for the task of diverse response generation. In contrast to the existing system and the baseline approaches, our proposed framework is capable of generating responses that are diverse in nature, thereby increasing the interactive property of the responses.

## 6.5 Human evaluation results

In Table 5, we present the human evaluation results for the baseline as well as the proposed approaches. The fluency of the proposed model is better in comparison to both the baseline models. We observe a fluency score of 41.31 for the proposed approach with stochastic beam search decoder (k = 5) along with BLOCK fusion techniques and attention modules. Hence, the models are capable of generating fluent responses. Also, there is an improvement of 4.98 in the relevance score over the baseline model. In case of diversity, it is evident from the table that the stochastic beam search decoder with k = 5 has achieved a score of 74.32, thereby, proving the fact that the responses generated are interesting and diverse in comparison to the baselines. Finally, the politeness score of the proposed approach is also better than the baseline model. Hence, the human evaluation results conclude that the proposed method has been successful in generating fluent, relevant, diverse as well as polite responses in a multi-modal dialogue system. In Fig 7, we present the responses generated by the different models. As it is

**Fig 7. Examples of responses generated by different models along with contextual information and input.**

visible from the figure, the responses generated are quite diverse and completely fluent. Also, instead of dull and direct Yes and No responses, the model has learned to be more polite by appreciating and apologizing, whenever required, according to the conversational history.

## 6.6 Attention analysis

For the model to be capable of learning different attribute information, we incorporate the knowledge base (KB) information in our model in the similar way as was done in [13]. To obtain a more focused information from KB, we apply self-attention on the KB inputs. The attention visualization of the KB is given in Fig 8. From the figure, it is noticeable that the model can focus on the relevant attributes for better dialogue generation. In Example 1 from the figure, it is evident that eyeglasses of brand Vincent is given more focus as the brand is an important feature for the user. Similarly, in Example 2 the print of the shirt is a significant aspect for the selection of appropriate shirt for the user. From these examples, it is clear that the KB input facilitates in capturing the correct attribute of a product, thereby, enhancing the quality of the generated response by making it more coherent to the user needs. Different attributes, such as the *brand, print, type, name* of a product gets attended by the self-attention applied on the KB input which is in accordance to the dialogue context as observed by the attention visualization in Fig 8 assists the generation of the textual responses.

For better interaction between the image and text, we employ parallel co-attention in our proposed framework as described in Section 3.3.1 of the Methodology section. For a complete understanding of the parallel co-attention and its effectiveness we provide an attention visualization of the parallel co-attention module in Fig 9. The attention on the text and the corresponding image is needed for better response generation. By employing parallel co-attention it

**Fig 8. Attention map for KB Input.**

## Example 1:
U1: What is the colour and material in the 4th result?
S1: It has grey colour cotton material.

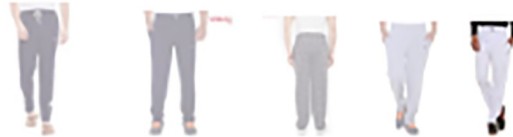

## Example 2:
U1: what material is it in the 2nd image?
S1: The brogues in the 2nd image has leather and rubber material

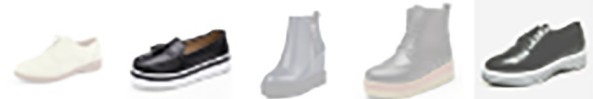

**Fig 9. Attention map for parallel co-attention between the text and image.**

can be observed from the attention visualization that the attention mechanism simultaneously focuses on the given text and the images, providing an enhanced representation to the decoder for the generation of better responses. From the figure, it is quite clear that the attributes and different images being discussed in text are also getting focused on the corresponding image. We thereby provide more information and better evidences to the decoder for generating the response. In the first example from the Figure, the 4[th] image gets attended as it is being discussed in the textual utterance as well. The color and material of the image and the information in the textual part of the utterance, such as cotton simultaneously gets attended providing improved representation of the entire utterance having both textual and visual features. Similarly, in Example 2 from the set of 5 images the 2[nd] image gets selected and the attributes such as leather, rubber material gets attended in the textual part of the utterance. From the figure showcasing the attention visualization, it can be concluded that the parallel co-attention mechanism assists in capturing the important and critical features of both the visual and textual part of the utterance simultaneously.

### 6.7 Error analysis

We closely analyze the outputs of the generated response so that we are aware of the errors made by the proposed dialogue generation framework. The common errors made by the model are:

- Additional Information: The model sometimes generates additional/extra information in case of attributes for a few products. For example, **Gold:** *The material of the trousers is cotton.*, **Predicted:** *The trousers have cotton polyester material with check patterns.*

- Incorrect information: The model is incapable of presenting the correct celebrity information in most of the cases. For example, **Gold:** *Celebrities cel_205, cel_2254 and cel_101 endorse this kind of shoes.*, **Predicted:** *This kind of shoes is endorsed by celebrity cel_123.*

- Erroneous image selection: The model is incompetent in selecting the images having contextual information of more than 5 turns, thereby generating incorrect responses in some cases. There are also the cases, where due to the discussion of multiple images in the conversational history, wrong images get selected, making the responses incorrect.

## 7 Conclusion and future work

In this paper, we have proposed a novel approach for diversifying textual responses in a multi-modal dialogue system. Our method makes use of parallel and dynamic attention to focus on image, text and knowledge base to capture the information present in both the modalities. BLOCK fusion technique was incorporated in the multi-modal encoder to obtain better representation from the modalities by improving the interaction between them. Experimental results prove that the attention mechanism and Block fusion technique help in generating correct and informative responses. For diversification, we have employed stochastic beam search with Gumble k-tricks. Detailed empirical analysis showsthat our proposed model is not only capable of generating informative responses, but also the responses are diverse and polite.

In future, we would take an opportunity of extending the architectural design to enhance the performance of our system. Also, we would focus on image retrieval and generation for building an end-to-end framework for multi-modal dialogue systems.

## Supporting information

**S1 File.**
(ZIP)

## Author Contributions

**Conceptualization:** Asif Ekbal.

**Data curation:** Mauajama Firdaus, Arunav Pratap Shandeelya.

**Funding acquisition:** Asif Ekbal.

**Investigation:** Mauajama Firdaus, Arunav Pratap Shandeelya, Asif Ekbal.

**Methodology:** Mauajama Firdaus, Arunav Pratap Shandeelya.

**Project administration:** Asif Ekbal.

**Software:** Arunav Pratap Shandeelya.

**Supervision:** Asif Ekbal.

**Validation:** Arunav Pratap Shandeelya.

**Writing – original draft:** Mauajama Firdaus.

**Writing – review & editing:** Asif Ekbal.

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
