## [Decision Letter · Decision Letter 0]

23 Jun 2020

PONE-D-20-02925

More to Diverse: Generating Diversified Responses in a Task Oriented Multimodal Dialog System

PLOS ONE

Dear Dr. Ekbal,

Thank you for submitting your manuscript to PLOS ONE. After careful consideration, we feel that it has merit but does not fully meet PLOS ONE’s publication criteria as it currently stands. Therefore, we invite you to submit a revised version of the manuscript that addresses the points raised during the review process.

We look forward to receiving your revised manuscript.

Kind regards,

Haoran Xie

Academic Editor

PLOS ONE

Journal Requirements:

'IMP/2018/002072 (http://www.imprint-2.in/Imprint-II/HomePage)'

Reviewers' comments:

Reviewer's Responses to Questions

**Comments to the Author**

1. Is the manuscript technically sound, and do the data support the conclusions?

Reviewer #1: No

Reviewer #2: Yes

2. Has the statistical analysis been performed appropriately and rigorously? 

Reviewer #1: No

Reviewer #2: Yes

3. Have the authors made all data underlying the findings in their manuscript fully available?

Reviewer #1: Yes

Reviewer #2: Yes

4. Is the manuscript presented in an intelligible fashion and written in standard English?

Reviewer #1: Yes

Reviewer #2: Yes

5. Review Comments to the Author

Reviewer #1: This research paper is based on the previously proposed multimodal system. Due to unimodal systems have got some problems in the dull and boring responses as authors indicated in the section 1.1 and 2.1, authors employed BLOCK fusion technique and used stochastic beam search with Gumble k-tricks to generate diversified responses (described in the section 2.2).

Authors have described the methods for proposed multimodal system. However, authors have also got a mention in the Editor’s Comment 3 that PLOS ONE requires experiments, statistics, and other analyses must be performed. I could not find any detail and illustration for the discussion of the section 5 Experiments. There is section 5.1 Implementation Details only. It is difficult to identify if this research is good or not.

Reviewer #2: In this paper, the authors propose a hierarchical encoder-decoder framework for generating diversified responses in a multimodal dialogue setup. It first uses the parallel co-attention to attend over the textual and visual contextual information simultaneously. Then, it applies the BLOCK fusion technique to improve multimodal representation learning. Finally, a stochastic beam search with Gumble Top K-tricks is applied to achieve diversified responses generation.

Strengths:

1) The structure, formulation, and figures are clear and straightforward.

2) The experimental design is sufficient; the ablative studies help in understanding different components in the model and the corresponding performance tradeoffs.

3) Better performance on the MMD dataset is demonstrated. It shows the efficacy of the proposed model.

Weaknesses:

1) The definition of the problem is confusing on page 5. U_{p,j}, and I_{p,j’}, what are the meanings of p, j, and j’? Why does history H_p just contain from (U_1, I_1) to （U_{p-1}, I_{p-1}）? What are the roles of the user U_p and image I_p? More important, as shown in Fig. 1, in the dialogue, S1 sometimes answer with both images and text. But in the paper, the authors just generate the text answer. The author should clarify the definition of the multimodal dialog task.

2) The authors have already introduced and discussed many works related to NLG. To my knowledge, the multimodal dialog task is much closer to the image dialog task. The references related to the dialog task are insufficient.

[1] Wu et al. Are you talking to me? Reasoned visual dialog generation through adversarial learning. In CVPR, 2018.

[2] Guo et al. Dual visual attention network for visual dialog. In IJCAI, 2019.

[3] Guo et al. Textual-Visual Reference-aware Attention Network for Visual Dialog. IEEE Transactions on Image Processing, 2020.

3) The novelty is limited. For example, parallel Co-Attention (PCA), including the BLOCK fusion for multimodal interaction, has been widely applied as a common attention framework. It is unclear about the motivation. Are there any more insights about your proposed model?

4) Fonts in Fig.6 are too small.

5) The implementation details of the politeness classifier in Fig.5 should be given in the paper.

6) Compared with [8] and [38], the performance superiority of the proposed method is inapparent. The authors should give more discussion.

6. PLOS authors have the option to publish the peer review history of their article (what does this mean?). If published, this will include your full peer review and any attached files.

Reviewer #1: No

Reviewer #2: No

---

## [Author Response · Author response to Decision Letter 0]

23 Jul 2020

Detailed response is attached at the end.

---

## [Decision Letter · Decision Letter 1]

2 Sep 2020

PONE-D-20-02925R1

More to Diverse: Generating Diversified Responses in a Task Oriented Multimodal Dialog System

PLOS ONE

Dear Dr. Ekbal,

Thank you for submitting your manuscript to PLOS ONE. After careful consideration, we feel that it has merit but does not fully meet PLOS ONE’s publication criteria as it currently stands. Therefore, we invite you to submit a revised version of the manuscript that addresses the points raised during the review process.

We look forward to receiving your revised manuscript.

Kind regards,

Haoran Xie

Academic Editor

PLOS ONE

Reviewers' comments:

Reviewer's Responses to Questions

**Comments to the Author**

1. If the authors have adequately addressed your comments raised in a previous round of review and you feel that this manuscript is now acceptable for publication, you may indicate that here to bypass the “Comments to the Author” section, enter your conflict of interest statement in the “Confidential to Editor” section, and submit your "Accept" recommendation.

Reviewer #1: (No Response)

Reviewer #2: All comments have been addressed

2. Is the manuscript technically sound, and do the data support the conclusions?

Reviewer #1: No

Reviewer #2: Yes

3. Has the statistical analysis been performed appropriately and rigorously? 

Reviewer #1: No

Reviewer #2: Yes

4. Have the authors made all data underlying the findings in their manuscript fully available?

Reviewer #1: Yes

Reviewer #2: Yes

5. Is the manuscript presented in an intelligible fashion and written in standard English?

Reviewer #1: Yes

Reviewer #2: Yes

6. Review Comments to the Author

Reviewer #1: Authors have applied the parallel co-attention technique between the textual utterance and the multiple images present to generate significant attributes and the textual response. Although authors have done the research in the multimodal dialogue system with the parallel co-attention technique, it is still necessary to give some experimental details for this article. In the Section 5.1, authors provided a brief explanation for the implementation details. There are still some key points that authors did not present for the implementation details of experiments about the determination of network parameters as follow：

(1). The AMSGrad was be used as the model training optimizer, and dropout was be used to avoid over-fitting that refer to Hinton's publication in 2014. Why the probability of 0.4 was be determined for the model training? Please give experimental details with the statistical analysis in the use of Table and Figure.

(2). In the same questions to determination of the fixed learning rate of 0.0004 and LSTM layers of 200, Please give experimental details with the statistical analysis in the use of Table and Figure.

Reviewer #2: The authors have adequately addressed my comments; the effort that the authors have made is appreciated. I would like to see this work for publication.

7. PLOS authors have the option to publish the peer review history of their article (what does this mean?). If published, this will include your full peer review and any attached files.

Reviewer #1: No

Reviewer #2: No

---

## [Author Response · Author response to Decision Letter 1]

14 Sep 2020

We appreciate the comments of the reviewers. In Section 6.3 (Analysis of Network Parameters) of the revised manuscript we have added the complete analysis with respect to the dropout, learning rate and LSTM layers for model fine-tuning. We used Perplexity (PPL) as the primary metric to fine-tune the framework for determining the network parameters in case of dropout and learning rate. After fixing these parameters we do the complete evaluation in terms of both automatic and human metrics for all the baselines and proposed framework. We used politeness accuracy (PA) as the primary metric for determining the number of LSTM layers in the politeness classifier. 

The dropout probability was determined to be 0.4 by considering the range of 0.1 - 0.8. The statistical analysis of the results for the different values of dropout probability for the proposed framework is provided in Table 3 of the revised manuscript. In Figure 4(a) we depict the performance of the proposed framework in case of dropout. It is evident that the proposed framework performs best when the dropout probability is 0.4. Therefore, for all the experiments (all baselines) we fix the dropout probability as stated. 

For learning rate we determined the value to be 0.0004, in a similar way as [12, 13, 44]. Also, we cross-verified by taking a range of 0.001 to 0.0001 to determine the learning rate in case of our proposed framework. As shown in Table 3 and Figure 6(b), the proposed framework performs best when the learning rate is fixed at 0.0004. This is in line with the existing literature. 

In the last paragraph of Section 6.3, we provide the analysis for the LSTM layers in the classifier. For the politeness classifier, we had fixed the LSTM layers at 200 by evaluating the performance of the classifier for the responses generated by our proposed framework. We checked the performance of the classifier by using different layers of LSTM as shown in Table 4 of the revised manuscript. From the Figure 6(c), it is evident that by increasing the number of LSTM layers the performance of the classifier improved with the maximum accuracy being 0.84 in case of 200 layers. On increasing the number of layers the performance declined and eventually became constant with accuracy lesser than the best performing classifier at 200 layers. This analysis helped in deciding the number of LSTM layers for the classifier.

---

## [Decision Letter · Decision Letter 2]

13 Oct 2020

More to Diverse: Generating Diversified Responses in a Task Oriented Multimodal Dialog System

PONE-D-20-02925R2

Dear Dr. Ekbal,

We’re pleased to inform you that your manuscript has been judged scientifically suitable for publication and will be formally accepted for publication once it meets all outstanding technical requirements.

Kind regards,

Haoran Xie

Academic Editor

PLOS ONE

Additional Editor Comments (optional):

Reviewers' comments:

Reviewer's Responses to Questions

**Comments to the Author**

1. If the authors have adequately addressed your comments raised in a previous round of review and you feel that this manuscript is now acceptable for publication, you may indicate that here to bypass the “Comments to the Author” section, enter your conflict of interest statement in the “Confidential to Editor” section, and submit your "Accept" recommendation.

Reviewer #1: All comments have been addressed

2. Is the manuscript technically sound, and do the data support the conclusions?

Reviewer #1: Yes

3. Has the statistical analysis been performed appropriately and rigorously? 

Reviewer #1: Yes

4. Have the authors made all data underlying the findings in their manuscript fully available?

Reviewer #1: Yes

5. Is the manuscript presented in an intelligible fashion and written in standard English?

Reviewer #1: Yes

6. Review Comments to the Author

Reviewer #1: (No Response)

7. PLOS authors have the option to publish the peer review history of their article (what does this mean?). If published, this will include your full peer review and any attached files.

Reviewer #1: No

---

## [Editor Report · Acceptance letter]

27 Oct 2020

PONE-D-20-02925R2 

More to Diverse: Generating Diversified Responses in a Task Oriented Multimodal Dialog System 

Dear Dr. Ekbal:

I'm pleased to inform you that your manuscript has been deemed suitable for publication in PLOS ONE. Congratulations! Your manuscript is now with our production department. 

Kind regards, 

on behalf of

Professor Haoran Xie 

Academic Editor

PLOS ONE